# On The Implicit Bias of Weight Decay in Shallow Univariate ReLU Networks

## Abstract

We give a complete characterization of the implicit bias of infinitesimal weight decay (i.e. an $\ell_2$ penalty on network weights) in the modest setting of univariate one layer ReLU networks. Our main result is a surprisingly simple geometric description of all one layer ReLU networks that exactly fit a dataset $\boldsymbol{D} = \{(x_i, y_i)\}$ with the minimum value of the $\ell_2$-norm of the neuron weights. Specifically, we prove that such functions must be either concave or convex between any two consecutive data sites $x_i$ and $x_{i+1}$. Our description implies that interpolating ReLU networks with weak $\ell_2$-regularization achieve the best possible $\ell_\infty$ generalization error for learning $1d$ Lipschitz functions, up to universal constants.

## 1 Introduction

The ability of overparameterized neural networks to simultaneously fit training data (i.e. interpolate) and generalize to unseen test data (i.e. extrapolate) is a robust empirical finding that underpins the success of deep learning in computer vision He et al. (2016); Krizhevsky et al. (2012), natural language processing Brown et al. (2020), and reinforcement learning Jumper et al. (2021); Silver et al. (2016); Vinyals et al. (2019). This observation is surprising when viewed from the lens of traditional learning theory Bartlett & Mendelson (2002); Vapnik & Chervonenkis (1971), chiefly because such complexity-based methods are agnostic to the choice of optimizer and seek to predict generalization based solely on the complexity of the overall hypothesis class and how well a learned model fits the training data.

In an overparameterized neural network, however, the quality of predictions at test time often varies dramatically across settings of trainable parameters (e.g. weights and biases) that exactly fit all training data Zhang et al. (2017). Which setting of parameters is learned depends crucially on the optimization procedure, and an insightful analysis of generalization in the presence of overparameterization must therefore combine properties of the model class with the often subtle criteria according to which different minimizers of an empirical risk are selected by different optimizers.

This has led to a vibrant sub-field of deep learning theory that analyzes the *implicit bias* or *implicit regularization* of optimizers used in practice Arora et al. (2019); Blanc et al. (2020); Gunasekar et al. (2018); Hanin & Sun (2021); Jacot et al. (2020); Ma et al. (2018); Razin & Cohen (2020); Smith et al. (2021). The high level goal of this line of work is to explain how optimization hyperparameters such as initialization scheme, learning rate, batch size, data augmentation scheme, and choice of explicit regularizer influence which of the many global minima of the empirical risk are selected in the course of optimization.

A key difficulty in studying implicit bias is that it is unclear how to understand, concretely in terms of the network function, the effect of particular optimization hyperparameters. For example, a well-chosen initialization for gradient-based optimizers is key to ensuring good generalization properties of the resulting learned network He et al. (2015); Mishkin & Matas (2015); Xiao et al. (2018). However, the corresponding geometric or analytic properties of the learned network are often hard to pin down, obscuring our understanding of what it is about the learned functions that encourages generalization.

In a similar vein, it is standard practice to experiment with explicit regularizers such as an $\ell_2$ penalty on network weights. While the effect of this choice is easy to describe in terms of model parameters (e.g. it tends to make them smaller), it is typically challenging to translate such a description into

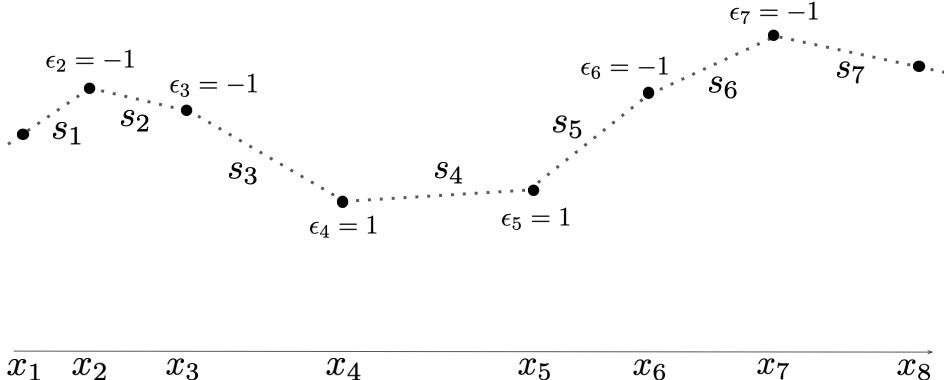

Figure 1: A dataset $\boldsymbol{D}$ with $m = 8$ points. Shown are the "connect the dots" interpolant $f_{\boldsymbol{D}}$ (dashed line), its slopes $s_i$ and the "discrete curvature" $\epsilon_i$ at each $x_i$.

properties of a learned non-linear model. In the simple setting of one layer ReLU networks there has been some relatively recent progress in this direction. Specifically, starting with an observation in Neyshabur et al. (2014) the articles Ongie et al. (2019); Parhi & Nowak (2020a;b; 2021); Savarese et al. (2019) explore and develop the fact that $\ell_2$ regularization on parameters in this setting is provably equivalent to penalizing the total variation of the derivative of the network function (cf eg Theorem 1.3 from prior work below). These articles apply to networks with any input dimension. In this article, however, we consider the simplest case of input dimension 1 and significantly refine these prior results to give a complete geometric answer to how interpolating ReLU networks with a weak $\ell_2$ penalty use training data to make predictions on unseen data. Our main results are:

1. We consider a dataset $\boldsymbol{D} = \{(x_i, y_i)\}$ with $x_i, y_i \in \mathbb{R}$ and give a complete description of the space of one layer ReLU networks with a single linear unit which fit the data and, among all such interpolating networks, do so with the minimal $\ell_2$ norm of the neuron weights. There are infinitely many such networks, and they are described by the constraint that they fit the data with as few inflection points as possible (see Thms. 1.1, 1.2).

2. The above description of the space of interpolants of $\boldsymbol{D}$ gives uniform control of the Lipschitz constant of any such interpolant and immediately yields sharp generalization bounds for learning $1d$ Lipschitz functions. This is stated in Corollary 1.1. Specifically, if the dataset $\boldsymbol{D}$ is generated by setting $y_i = f_*(x_i)$ for $f_* : [0, 1] \to \mathbb{R}$ a Lipschitz function, then any one layer ReLU network with a single linear unit which interpolates $\boldsymbol{D}$ but does so with minimal $\ell_2$-norm of the network parameters will generalize as well as possible to unseen data, up to a small universal multiplicative constant. To the author's knowledge this is the first time such generalization guarantees have been obtained.

## 1.1 SETUP AND INFORMAL STATEMENT OF RESULTS

Let us denote
$$[t]_+ := \mathrm{ReLU}(t) = \max\{0, t\}$$
and consider a one layer ReLU network

$$z(x) = z(x; \theta) = z(x; \theta, n) := ax + b + \sum_{j=1}^{n} W_j^{(2)} \left[W_j^{(1)}x + b_i^{(1)}\right]_+ \qquad (1)$$

with input and output dimensions equal to 1 and a single linear unit[1] and . For a given dataset

$$\boldsymbol{D} = \{(x_i, y_i), \, i = 1, \ldots, m\}, \qquad -\infty < x_1 < \cdots < x_m < \infty, \quad y_i \in \mathbb{R},$$

if the number of datapoints $m$ is smaller than the network width $n$, there are infinitely many choices of the parameter vector $\theta$ for which $z(x; \theta)$ interpolates (i.e. fits) the data:

$$z(x_i; \theta) = y_i, \qquad \forall \, i = 1, \ldots, m. \tag{2}$$

Without further information about how $\theta$ was selected, little can be said about the function $x \mapsto z(x; \theta)$ on intervals $(x_i, x_{i+1})$ between two consecutive datapoints when $n$ is much larger than $m$. This precludes useful generalization guarantees that hold uniformly over all $\theta$ subject only to the interpolation condition equation 2.

In practice interpolants are not chosen arbitrary. Instead, they are typically learned by some variant of gradient descent starting from a random initialization. For a given network architecture, initialization scheme, optimizer, data augmentation scheme, regularizer, and so on, understanding how the learned network uses the known labels $\{y_i, \, i = 1, \ldots, m\}$ to extrapolate values of $z(x; \theta)$ for $x$ in intervals $(x_i, x_{i+1})$ away from the datapoints in $\boldsymbol{D}$ is an important open problem. To obtain non-trivial generalization estimates and make progress on this problem, a fruitful line of inquiry in prior work has been to search for additional complexity measures based on margins Wei et al. (2018), PAC-Bayes estimates Dziugaite & Roy (2017; 2018); Nagarajan & Kolter (2019), weight matrix norms Bartlett et al. (2017); Neyshabur et al. (2015), information theoretic compression estimates Arora et al. (2018), Rachemacher complexity Golowich et al. (2018), etc that, while perhaps not explicitly regularized, are hopefully small in trained networks. The idea is then that these complexity measures being small gives additional constrains on the capacity of the space of learned networks. We refer the interested reader to Jiang et al. (2019) for a review and empirically comparison of many such approaches.

In this article, we take a different approach to studying generalization. We do not seek general results that are valid for any network architecture. Instead, our goal is to describe completely, in concrete geometrical terms, the properties of one layer ReLU networks $z(x; \theta)$ that interpolate a dataset $\boldsymbol{D}$ in the sense of equation 2 with the minimal possible $\ell_2$ penalty

$$C(\theta) = C(\theta, n) = \frac{1}{2} \sum_{j=1}^{n} \left( \left| W_j^{(1)} \right|^2 + \left| W_j^{(2)} \right|^2 \right)$$

on the neuron weights. More precisely, we study the space of ridgeless ReLU interpolants $\mathrm{RidgelessReLU}(\boldsymbol{D})$ of a dataset $\boldsymbol{D}$, defined by

$$\left\{ f : \mathbb{R} \to \mathbb{R} \mid \exists \theta, n \text{ s.t. } f(x) = z(x; \theta) \, \forall x \in \mathbb{R}, \, z(x_i; \theta) = y_i \, \forall i = 1, \ldots, m, \, C(\theta) = C_* \right\}, \tag{3}$$

where

$$C_* := \inf_{\theta, n} \left\{ C(\theta, n) \mid z(x_i; n, \theta) = y_i \ \forall (x_i, y_i) \in \boldsymbol{D} \right\}.$$

While we do not prove this directly here, a simple intuition for the elements of $\mathrm{RidgelessReLU}(\boldsymbol{D})$ is that they are all univariate one layer ReLU networks that minimize a weakly penalized loss

$$\boldsymbol{L}(\theta; \boldsymbol{D}) + \lambda C(\theta), \qquad \lambda \ll 1, \tag{4}$$

where $\boldsymbol{L}$ is an empirical loss, such as the mean squared error over $\boldsymbol{D}$, and the strength $\lambda$ of the weight decay penalty $C(\theta)$ is infinitesimal. There is an important subtlety in the definition of $\mathrm{RidgelessReLU}(\boldsymbol{D})$. Namely, given $\theta$, there exist infinitely many $\widetilde{\theta}$ such that $z(x; \theta) = z(x; \widetilde{\theta})$ for every $x$. Thus, a function $f$ belongs to $\mathrm{RidgelessReLU}(\boldsymbol{D})$ if $f$ interpolates the dataset $\boldsymbol{D}$ and $z(x; \theta) = f(x)$ for some setting of $\theta$ that achieves the minimal value of $C(\theta)$ among all such interpolants.

It it plausible but by no means obvious that, with high probability, gradient descent from a random initialization and a weight decay penalty whose strength decreases to zero over training converges to an element in $\mathrm{RidgelessReLU}(\boldsymbol{D})$. This article does not study optimization, and we therefore leave this as an interesting open problem. Our main result is simple description of $\mathrm{RidgelessReLU}(\boldsymbol{D})$ and can informally be stated as follows:

---

[1]The presence of the linear term $ax + b$ is not really standard in practice but is adopted in keeping with prior work Ongie et al. (2019); Parhi & Nowak (2020a); Savarese et al. (2019) since it leads a cleaner mathematical formulation of results.

**Theorem 1.1** (Informal Statement of Theorem 1.2)**.** *Fix a dataset $\boldsymbol{D} = \{(x_i, y_i), \ i = 1, \ldots, m\}$ and define*

$$\epsilon_i := \text{sgn}\left(s_i - s_{i-1}\right), \qquad s_i := \frac{y_{i+1} - y_i}{x_{i+1} - x_i}.$$

*Note that $s_i$ is the slope of the line connecting $(x_i, y_i)$ to $(x_{i+1}, y_{i+1})$ and that $\epsilon_i$ is an estimate for the sign of the local curvature of the function that generated the data (Figure 1). Among all continuous and piecewise linear functions $f$ that interpolate $\boldsymbol{D}$ exactly, the ones in $\text{RidgelessReLU}(\boldsymbol{D})$ are precisely those that:*

- *Are linear (or more precisely affine) on intervals $(x_i, x_{i+1})$ when neighboring datapoints disagree on the local curvature in the sense that $\epsilon_i \cdot \epsilon_{i+1} \neq 1$.*

- *Are convex (resp. concave) on sequences of intervals $(x_i, x_{i+1}), \ldots, (x_{i+q-1}, x_{i+q})$ on which datapoints $x_i, \ldots, x_{i+q}$ agree on the local curvature in the sense that $\epsilon_i = \cdots = \epsilon_{i+q} = 1$ (resp. $\epsilon_i = \epsilon_{i+1} = -1$). On such intervals $f$ lies below (resp. above) the straight line interpolant of the data. See Figures 5 and 7.*

Before giving a precise statement our results, we mention that, as described in detail below, the space $\text{RidgelessReLU}(\boldsymbol{D})$ has been considered in a number of prior articles Ongie et al. (2019); Parhi & Nowak (2020a); Savarese et al. (2019). Our starting point will be the useful but abstract characterization of $\text{RidgelessReLU}(\boldsymbol{D})$ they obtained in terms of the total variation of the derivative of $z(x; \theta)$ (see equation 5).

Let us also note that the conclusions of Theorem 1.1 (and Theorem 1.2) also hold under seemingly very different hypotheses from ours. Namely, instead of $\ell_2$-regularization on the parameters, Blanc et al. (2020) considers SGD training for mean squared error with iid noise added to labels. Their Theorem 2 shows (modulo some assumptions about interpreting the derivative of the ReLU) that, among all ReLU networks a linear unit that interpolate a dataset $\boldsymbol{D}$, the only ones that minimize the implicit regularization induced by adding iid noise to SGD are precisely those that satisfy the conclusions of Theorem 1.1 and hence are exactly the networks in $\text{RidgelessReLU}(\boldsymbol{D})$. This suggests that our results hold under much more general conditions. It would be interesting to characterize them.

Further, our characterization of $\text{RidgelessReLU}(\boldsymbol{D})$ in Theorem 1.2 immediately implies strong generalization guarantees uniformly over $\text{RidgelessReLU}(\boldsymbol{D})$. We give a representative example in Corollary 1.1, which shows that such ReLU networks achieve the best possible generalization error of Lipschitz functions, up to constants.

Finally, note that we allow networks $z(x; \theta)$ of any width but that if the width $n$ is too small relative to the dataset size $m$, then the interpolation condition equation 2 cannot be satisfied. Also, we point out that in our formulation of the cost $C(\theta)$ we have left both the linear term $ax + b$ and the neuron biases unregularized. This is not standard practice but seems to yield the cleanest results.

## 1.2 STATEMENT OF RESULTS AND RELATION TO PRIOR WORK

Every ReLU network $z(x; \theta)$ is a continuous and piecewise linear function from $\mathbb{R}$ to $\mathbb{R}$ with a finite number of affine pieces. Let us denote by $\text{PL}$ the space of all such functions and define

$$\text{PL}(\boldsymbol{D}) := \{f \in \text{PL} \mid f(x_i) = y_i \ \forall i = 1, \ldots, m\}$$

to be the space of piecewise linear interpolants of $\boldsymbol{D}$. Perhaps the most natural element in $\text{PL}(\boldsymbol{D})$ is the "connect-the-dots interpolant" $f_{\boldsymbol{D}} : \mathbb{R} \to \mathbb{R}$ given by

$$f_{\boldsymbol{D}}(x) := \begin{cases} \ell_1(x), & x < x_2 \\ \ell_i(x), & x_i < x < x_{i+1}, \quad i = 2, \ldots, m-2 \ , \\ \ell_{m-1}(x), & x > x_{m-1} \end{cases}$$

where for $i = 1, \ldots, m-1$, we've set

$$\ell_i(x) := (x - x_i)s_i + y_i, \qquad s_i := \frac{y_{i+1} - y_i}{x_{i+1} - x_i}.$$

See Figure 1. In addition to $f_{\boldsymbol{D}}$, there are many other elements in $\mathrm{RidgelessReLU}(\boldsymbol{D})$. Theorem 1.2 gives a complete description of all of them phrased in terms of how they may behave on intervals $(x_i, x_{i+1})$ between consecutive datapoints. Our description is based on the signs

$$\epsilon_i = \mathrm{sgn}\left(s_i - s_{i-1}\right), \qquad 2 \le i \le m$$

of the (discrete) second derivatives of $f_{\boldsymbol{D}}$ at the inputs $x_i$ from our dataset.

**Theorem 1.2.** *The space* $\mathrm{RidgelessReLU}(\boldsymbol{D})$ *consists of those* $f \in \mathrm{PL(D)}$ *satisfying:*

1. *$f$ coincides with $f_{\boldsymbol{D}}$ on the following intervals:*

    *(1a) Near infinity, i.e. on the intervals $(-\infty, x_2), (x_{m-1}, \infty)$*

    *(1b) Near datapoints that have zero discrete curvature, i.e. on intervals $(x_{i-1}, x_{i+1})$ with $i = 2, \ldots, m - 1$ such that $\epsilon_i = 0$.*

    *(1c) Between datapoints with opposite discrete curvature, i.e. on intervals $(x_i, x_{i+1})$ with $i = 2, \ldots, m - 1$ such that $\epsilon_i \cdot \epsilon_{i+1} = -1$.*

2. *$f$ is convex (resp. concave) and bounded above (resp. below) by $f_{\boldsymbol{D}}$ between any consecutive datapoints at which the discrete curvature is positive (resp. negative). Specifically, suppose for some $3 \le i \le i + q \le m - 2$ that $x_i$ and $x_{i+q}$ are consecutive discrete inflection points in the sense that*

$$\epsilon_{i-1} \ne \epsilon_i, \qquad \epsilon_i = \cdots = \epsilon_{i+q}, \qquad \epsilon_{i+q} \ne \epsilon_{i+q+1}.$$

   *If $\epsilon_i = 1$ (resp. $\epsilon_i = -1$), then restricted to the interval $(x_i, x_{i+q})$, $f$ is convex (resp. concave) and lies above (resp. below) the incoming and outgoing support lines and below (resp. above) $f_{\boldsymbol{D}}$:*

$$\epsilon_i = 1 \qquad \Longrightarrow \qquad \max\left\{\ell_{i-1}(x),\, \ell_{i+q}(x)\right\} \ \le \ f(x) \le f_{\boldsymbol{D}}(x)$$
$$\epsilon_i = -1 \qquad \Longrightarrow \qquad \min\left\{\ell_{i-1}(x),\, \ell_{i+q}(x)\right\} \ \ge \ f(x) \ \ge \ f_{\boldsymbol{D}}(x)$$

   *for all $x \in (x_i, x_{i+q})$.*

We prove Theorem 1.2 in §A. Before doing so, let us illustrate Theorem 1.2 as an algorithm that, given the dataset $\boldsymbol{D}$, describes all elements in $\mathrm{RidgelessReLU}(\boldsymbol{D})$ (see Figures 5 and 7):

Step 1 **Linearly interpolate the endpoints**: by property (1), $f \in \mathrm{RidgelessReLU}(\boldsymbol{D})$ must agree with $f_{\boldsymbol{D}}$ on $(-\infty, x_2)$ and $(x_{m-1}, \infty)$.

Step 2 **Compute discrete curvature**: for $i = 2, \ldots, m - 1$ calculate the discrete curvature $\epsilon_i$ at the data point $x_i$.

Step 3 **Linearly interpolate on intervals with zero curvature**: for all $i = 2, \ldots, m - 1$ at which $\epsilon_i = 0$ property (1) guarantees that $f$ coincides with the $f_{\boldsymbol{D}}$ on $(x_{i-1}, x_{i+1})$.

Step 4 **Linearly interpolate on intervals with ambiguous curvature**: for all $i = 2, \ldots, m - 1$ at which $\epsilon_i \cdot \epsilon_{i+1} = -1$ property (1) guarantees that $f$ coincides with $f_{\boldsymbol{D}}$ on $(x_i, x_{i+1})$.

Step 5 **Determine convexity/concavity on remaining points**: all intervals $(x_i, x_{i+1})$ on which $f$ has not yet been determined occur in sequences $(x_i, x_{i+1}), \ldots, (x_{i+q-1}, x_{i+q})$ on which $\epsilon_{i+j} = 1$ or $\epsilon_{i+j} = 1$ for all $j = 0, \ldots, q$. If $\epsilon_i = 1$ (resp. $\epsilon_i = -1$), then $f$ is any convex (resp. concave) function bounded below (resp. above) by $f_{\boldsymbol{D}}$ and above (resp. below) the support lines $\ell_i(x)$, $\ell_{i+q}(x)$.

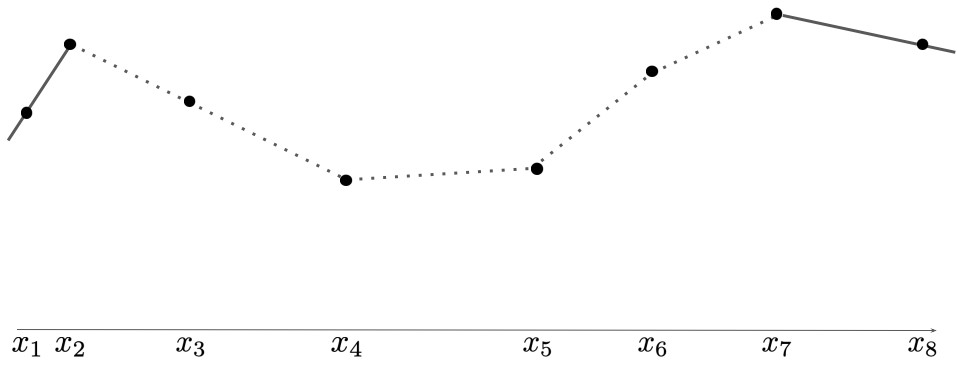

Figure 2: Step 1

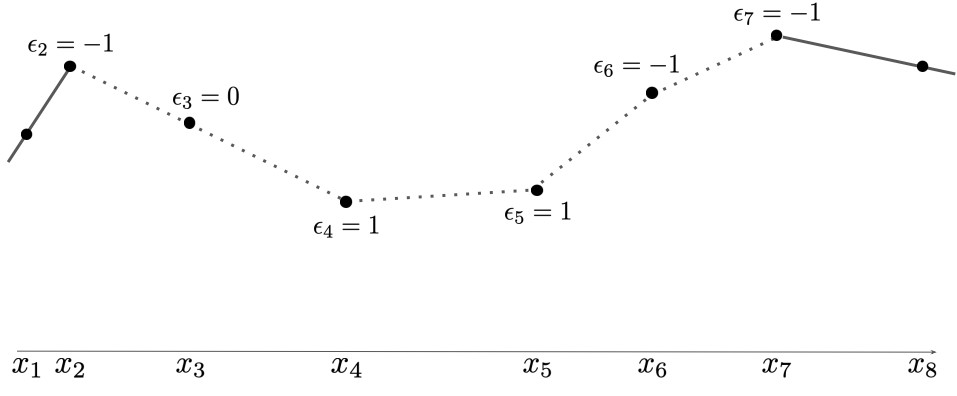

Figure 3: Step 2

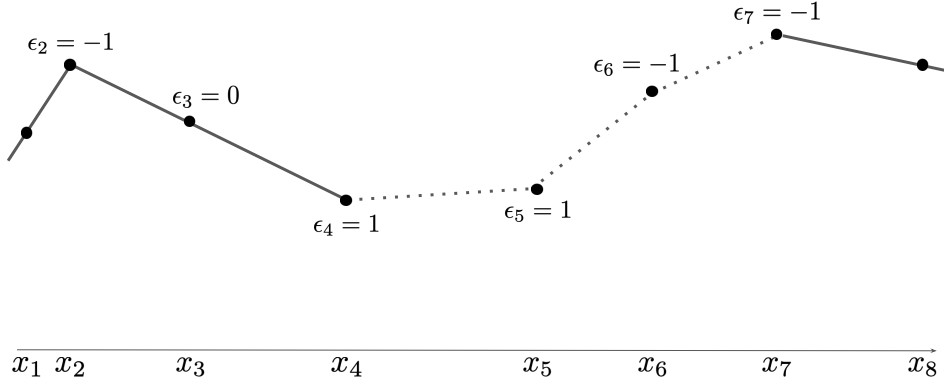

Figure 4: Step 3

Figure 5: Steps 1 - 3 for generating $\mathrm{RidgelessReLU}(\boldsymbol{D})$ from the dataset $\boldsymbol{D}$.

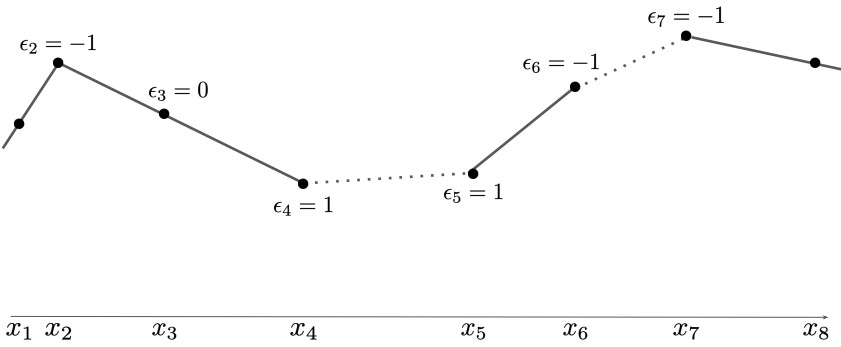

Figure 6: Step 4

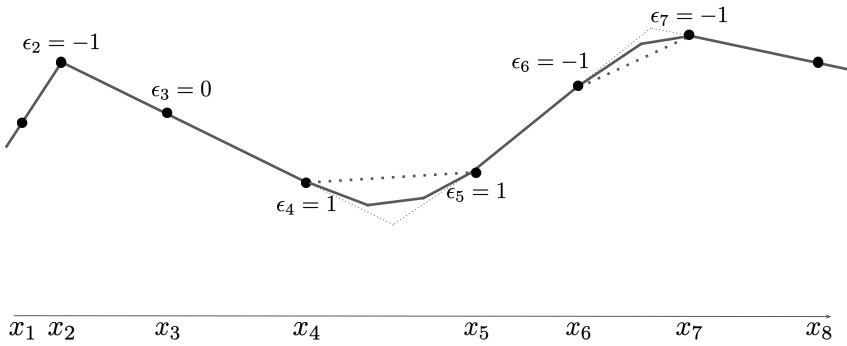

Figure 7: Step 5. One possible choice of a convex interpolant on $(x_4, x_5)$ and of a concave interpolant on $(x_6, x_7)$ is shown. Thin dashed lines are the supporting lines that bound all interpolants below on $(x_4, x_5)$ and above on $(x_6, x_7)$.

The starting point for the proof of Theorem 1.2 comes from the prior articles Neyshabur et al. (2014); Ongie et al. (2019); Savarese et al. (2019), which obtained an insightful "function space" interpretation of RidgelessReLU($\boldsymbol{D}$) as a subset of PL($\boldsymbol{D}$). Specifically, a simple computation (cf e.g. Theorem 3.3 in Savarese et al. (2019) and also Lemma A.4 below) shows that $f_{\boldsymbol{D}}$ achieves the smallest value of the total variation $||Df||_{TV}$ for the derivative $Df$ among all $f \in$ PL($\boldsymbol{D}$). (The function $Df$ is piecewise constant and $||D\tilde{f}||_{TV}$ is the sum of absolute values of its jumps.) Part of the content of the prior work Neyshabur et al. (2014); Ongie et al. (2019); Savarese et al. (2019) is the following result

**Theorem 1.3** (cf Lemma 1 in Ongie et al. (2019) and near eq. (17) Savarese et al. (2019) ). *For any dataset $\boldsymbol{D}$ we have*

$$\text{RidgelessReLU}(\boldsymbol{D}) = \{f \in \text{PL}(\boldsymbol{D}) \mid ||Df||_{TV} = ||Df_{\boldsymbol{D}}||_{TV}\}. \tag{5}$$

Theorem 1.3 says that RidgelessReLU($\boldsymbol{D}$) is precisely the space of functions in PL($\boldsymbol{D}$) that achieve the minimal possible total variation norm for the derivative. Intuitively, functions in RidgelessReLU($\boldsymbol{D}$) are therefore averse to oscillation in their slopes. The proof of this fact uses a simple idea introduced in Theorem 1 of Neyshabur et al. (2014) which leverages the homogeneity of the ReLU to translate between the regularizer $C(\theta)$, which is positively homogeneous of degree 2 in the network weights, and the penalty $||Df||_{TV}$, which is positively homogeneous of degree 1 in the network function.

Theorem 1.2 yields strong generalization guarantees uniformly over RidgelessReLU($\boldsymbol{D}$). To state a representative example, suppose $\boldsymbol{D}$ is generated by a function $f_* : [0, 1] \to \mathbb{R}$:

$$y_j = f_*(x_j).$$

We then find the following

**Corollary 1.1** (Sharp generalization on Lipschitz functions over a compact set). *Fix a dataset $\boldsymbol{D} = \{(x_i, y_i),\ i = 1, \ldots, m\}$ with $x_i \in [0, 1]$. We have*

$$\sup_{f \in \text{RidgelessReLU}(\boldsymbol{D})} ||f||_{\text{Lip}} \le ||f_*||_{\text{Lip}}. \tag{6}$$

*Hence, if $f_*$ is $L-$Lipschitz and we denote by $\Delta := \max_{i=0}^{m+1} \min_{j \ne i} \{x_i - x_j\}$ the maximal distance between consecutive training points (with $x_0 = 0, x_{m+1} = 1$), then*

$$\sup_{f \in \text{RidgelessReLU}(\boldsymbol{D})} \sup_{x \in [0,1]} |f(x) - f_*(x)| \le \Delta L, \tag{7}$$

*which is the best generalization error possible, up to multiplicative constants.*

*Proof.* Observe that for any $i = 2, \ldots, m - 1$ and $x \in (x_i, x_{i+1})$ at which $Df(x)$ exists we have

$$\epsilon_i(s_{i-1} - s_i) \le \epsilon_i(Df(x) - s_i) \le \epsilon_i(s_{i+1} - s_i). \tag{8}$$

Indeed, when $\epsilon_i = 0$ the estimate equation 8 follows from property (1b) in Theorem 1.2. Otherwise, equation 8 follows immediately from the local convexity/concavity of $f$ in property (2). Hence, combining equation 8 with property (1a) shows that for each $i = 1, \ldots, m - 1$

$$||Df||_{L^\infty(x_i, x_{i+1})} \le \max\{|s_{i-1}|, |s_i|\}.$$

Again using property (1a) and taking the maximum over $i = 2, \ldots, m$ we find

$$||Df||_{L^\infty(\mathbb{R})} \le \max_{1 \le i \le m-1} |s_i| = ||f_{\boldsymbol{D}}||_{\text{Lip}}.$$

To complete the proof of equation 6 observe that for every $i = 1, \ldots, m - 1$

$$|s_i| = \left|\frac{y_{i+1} - y_i}{x_{i+1} - x_i}\right| = \left|\frac{f_*(x_{i+1}) - f_*(x_i)}{x_{i+1} - x_i}\right| \le ||f_*||_{\text{Lip}} \implies ||f_{\boldsymbol{D}}||_{\text{Lip}} \le ||f_*||_{\text{Lip}}.$$

Given any $x \in [0, 1]$, let us write $x'$ for its nearest neighbor in $\{x_i,\ i = 0, \ldots, m + 1\}$. We find

$$|f(x) - f_*(x)| \le |f(x) - f(x')| + |f_*(x') - f_*(x)| \le \left(||f||_{\text{Lip}} + ||f_*||_{\text{Lip}}\right)|x - x'| \le L\Delta.$$

Taking the supremum over $f \in$ RidgelessReLU($\boldsymbol{D}$) and $x \in [0, 1]$ proves equation 7. $\qquad\square$

Corollary 1.1 gives the best possible generalization error of Lipschitz functions, up to a universal multiplicative constant, in the sense that if all we knew about $f_* : [0, 1] \to \mathbb{R}$ was that it was $L$-Lipschitz and were given its values on $\{x_i, i = 1, \ldots, m\}$, then we cannot recover $f_*$ in $L^\infty$ to accuracy that is better than a constant times $L\Delta$. For $m$ uniformly spaced points we have $\Delta = 1/m + 1$, while classical results (e.g. Theorem 2.2. in Holst (1980)) show that if $x_i \sim \text{Unif}([0, 1])$ are iid, then $\Delta$ is bounded above by a constant time $\log(m)/m$ with high probability.

## 1.3 OUTLINE OF PROOF OF THEOREM 1.2

In this section, we briefly outline the main steps in proving Theorem 1.2:

- A "local straightening" result given in Proposition A.1. This shows that any element $f$ in $\text{RidgelessReLU}(\boldsymbol{D})$ be either convex or concave on any interval of the form $(x_i, x_{i+1})$ between two consecutive inputs in the training data. The main idea is that non-monotonicity of $Df$ on such intervals can only increase $||Df||_{TV}$.

- A "linearity at endpoints" result given in Proposition A.3. This shows that any element $f \in \text{RidgelessReLU}(\boldsymbol{D})$ agrees with $f_{\boldsymbol{D}}$ to the left of $x_2$ and to the right of $x_{m-1}$. The main idea is that, given $f$ restricted to $(x_2, x_{m-1})$, a linear extension of $f$ to the complement of this interval can already interpolate the values at $x_1, x_m$ at zero additional cost to $||Df||_{TV}$.

- A "left-right compatibility" result given in Propositions A.4, A.5, A.6. This gives constraints, by dividing into cases, on the monotonicity of "incoming slopes" $s_{\text{in}}(x_i)$ and "outgoing slopes" $s_{\text{out}}(x_i)$ of any $f \in \text{RidgelessReLU}(\boldsymbol{D})$. The main idea is that the slope of $f$ on each interval $(x_i, x_{i+1})$ must attain values that are both less than or equal and great than or equal to the slope $s_i$ of the $f_{\boldsymbol{D}}$. This give constraints between $s_{i-1}, s_i, s_{\text{in}}(x_i), s_{\text{out}}(x_i)$.

- Combining the preceding results allows us to conclude that $\text{RidgelessReLU}(\boldsymbol{D})$ is a subset of the set of functions satisfying the conclusions of Theorem 1.2.

- Finally, Proposition A.7 shows that the set of functions satisfying the conclusions of Theorem 1.2 are a subset of $\text{RidgelessReLU}(\boldsymbol{D})$.

## 1.4 DISCUSSION OF LIMITATIONS AND FUTURE WORK

In this article, we completely characterized all possible ReLU networks that interpolate a given dataset $\boldsymbol{D}$ in the simple setting of weakly $\ell_2$-regularized one layer ReLU networks with a single linear unit and input/output dimension 1. Moreover, our characterization shows that, to assign labels to unseen data such networks simply "look at the curvature of the nearest neighboring datapoints on each side," in a way made precise in Theorem 1.2. This simple geometric description led to sharp generalization results for learning $1d$ Lipschitz functions in Corollary 1.1.

This opens many direction for future investigation. Theorem 1.2 shows, for instance, that there are infinitely many ridgeless ReLU interpolants of a given dataset $\boldsymbol{D}$. It would be interesting to understand which ones are actually learned by gradient descent from a random initialization and a weak (or even decaying) $\ell_2$-penalty in time. Further, as already pointed out after the Theorem 1.1, the conclusions of Theorem 1.2 appear to hold under very different kinds of regularization (e.g. Theorem 2 in Blanc et al. (2020)). This raises the question: what is the most general kind of regularizer that is equivalent to weight decay, at least in our simple setup?

Finally, it would also be quite natural to extend the results in this article to ReLU networks with higher input dimension, for which weight decay is known to correspond to regularization of a certain weighted Radon transform of the network function Ongie et al. (2019); Parhi & Nowak (2020a;b; 2021). Finally, extending the results in this article to deeper networks and beyond fully connected architectures are directions left to future work.

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
