# OpenReview forum: "On The Implicit Bias of Weight Decay in Shallow Univariate ReLU Networks"
_ICLR.cc/2023/Conference — Submitted to ICLR 2023_

### Official Review · Reviewer_oSFb · 2022-10-24

**Confidence:** 3
**Correctness:** 4
**Technical Novelty And Significance:** 3
**Empirical Novelty And Significance:** 3
**Recommendation:** 8

**Clarity, Quality, Novelty And Reproducibility:**

Clarity: The paper is written clearly.
Quality: I think this is a good paper.
Novelty: The geometric characterisation is novel.
Reproducibility: This is not an empirical paper.


**Strength And Weaknesses:**

Weaknesses:
1. The paper does not study optimization, only the generalisation guarantees that are achieved by the minimiser.
2. Only the one-dimensional setting is studied.

Strengths:
1. The authors describe an algorithm which is a consequence of the theorem, that allows them to describe all elements that might minimise the $\ell_2$ norm of the weights while interpolating the dataset.
2. I personally like the attempt at a geometric characterisation of the min-cost interpolating function, and generalisation guarantees that might arise from that.


**Summary Of The Paper:**

This paper attempts to study the generalisation properties of one-dimensional ReLU networks directly, instead of via the usual VC-dimension or Rademacher complexity-type bounds.

To this end, they *geometrically* characterise the space of one-layer ReLU networks with a single linear unit, which *fit* a given dataset $D \subset \mathbb{R}^2$ *while achieving the minimum possible $\ell_2$ norm among the neuron weights*.

Main Results:
A geometric characterisation of all one-layer ReLU networks that exactly fit a univariate dataset $D \subset \mathbb R^2$. Since ReLU networks in this context are just piecewise linear functions, this turns out to be described by the constraint that the network fits the data while achieving as few inflection points as possible.

Corollary: Interpolating ReLU networks with $\ell_2$ regularisation achieve best possible $\ell_\infty$ generalisation error for one-dimensional Lipschitz functions.

Their work builds on the observation due to Neyshabur(2014) that $\ell_2$ regularisation in the setting considered is equivalent to penalising the TV distance of the derivative of the network function. This observation applies to any dimension, but the complete characterisation in this paper is only for one-dimensional neural networks.


**Summary Of The Review:**

This paper gives a complete characterisation of the $\ell_2$ minimizing univariate ReLU network.

I think this is a complete result for a simple setting that provides interesting insights into the generalisation as well as geometric properties of the minimiser, and vote to accept it.

---

> ### Author Response · Authors · 2022-11-07
> **Thank you**
>
> We thank the reviewer for their careful reading and positive assessment of our paper. We agree with the reviewer's summary of the weaknesses in the article. We are certainly working to overcome both the 1D input assumption and the lack of optimization guarantees in subsequent work!

---

### Official Review · Reviewer_LdBk · 2022-10-25

**Confidence:** 3
**Correctness:** 2
**Technical Novelty And Significance:** 1
**Empirical Novelty And Significance:** 1
**Recommendation:** 3

**Clarity, Quality, Novelty And Reproducibility:**

The setup is too restrictive (e.g., needs 1-D input) and not justified properly. The following are some detailed comments.

1. Minimizing the l_2-norm of all neuron weights is not weight decay. The authors confirm this at the end of Page~3. While the authors argue that they do not study optimization and do not care whether such a setup has any relationship with the actual gradient descent with weight decay, the title of this paper still highlights the weight decay, which in my opinion is totally misleading and confusing.

2. The requirement of Corollary 1.1 is restrictive, as it needs all training inputs uniformly spaced (deterministically) in [0, 1]. Although the authors argue that a similar result also holds if the 1-D input is drawn independently at random from [0, 1], no rigorous proof is provided and thus such a claim is less convincing.

3. Figs. 1~7 take too much space (almost 2.5 pages) and are repetitive. No simulations on the test error are provided.

**Strength And Weaknesses:**

Strength: The presentation is relatively clear.
Weakness: The setup is too restrictive and not justified properly. See my comments below.

**Summary Of The Paper:**

This paper considers a one-layer ReLU network that only considers 1-D input. The authors study a set of solutions RidgelessReLU(D) that has the smallest norm of weights among all interpolators of the training samples. The authors show that the functions in such a set satisfy certain requirements by examining the second-order discrete derivative. Based on these requirements, this paper provides a generalization upper bound on Lipschitz functions when the 1-D input is uniformly spaced (deterministically) in [0, 1], and argues that a similar result also holds if the 1-D input is drawn independently at random from [0, 1].

**Summary Of The Review:**

I don't think this paper will be helpful in understanding the performance of existing neural networks, because of 1) an oversimplified setup, and 2) no connection/similarity with the actual gradient descent with weight decay used in the neural network.

---

> ### Author Response · Authors · 2022-11-07
> **Thank you + Response to Comments**
>
> ### Summary and Reply
>
> We thank the reviewer for their review, which was largely negative in light of three concerns:
>
> * The issue that weight decay and $\ell_2$-regularization are not the same
> * The hypotheses of Corollary 1.1
> * The lack of simulations
>
> ***We respectfully invite the reviewer to comment on whether the following will address their primary concerns:***
>
> * We will rephrase Corollary 1.1 directly in terms of the maximal spacing $\Delta$ between training datapoints. With an identical proof, the result will then read that the generalization error on Lip 1 functions $f:[0,1]\rightarrow \mathbb R$ scales like $2\Delta$, which is optimal aside from the pre-factor. In particular, we would like to indicate to the reviewer our strong disagreement with their negative assessment of the generality of this result. To the best of our knowledge, **this is the first generalization guarantee on Lip 1 functions for even mildly realistic neural networks.**
>
> * While the reviewer is certainly correct that in general there is a different between weight decay and $\ell_2$-regularization, we note that they are the same for GD and SGD and are therefore often used interchangeably. Regardless, we will add in the revision a sentence to the abstract and beginning on introduction making this clear.
>
> * We do not believe that article would be significantly helped by simulations, beyond those already present in the prior work by Savarese et. al. and Ongie et. al. These prior articles, whose empirics we will explicitly point to in the revision, check that optimization with a small $\ell_2$ penalty indeed gets close to the minimal norm interpolant.

---

### Official Review · Reviewer_Y8Pp · 2022-10-25

**Confidence:** 4
**Correctness:** 4
**Technical Novelty And Significance:** 2
**Empirical Novelty And Significance:** Not applicable
**Recommendation:** 5

**Clarity, Quality, Novelty And Reproducibility:**

The paper is well written. The assumptions and results are clearly stated, and the results given in the paper are new, to the best of my knowledge.

**Strength And Weaknesses:**

**Strengths:**
- The paper is well written and explains clearly how the results relate to previous works.
- The generalization guarantee for ReLU networks is an interesting and novel result, to the best of my knowledge.

**Weaknesses:**
* The main weakness is that the setting is quite restricted. The results only apply to univariate one hidden layer ReLU networks which interpolate the training data and minimize a $\ell_2$ penalty. As the authors mention, the penalty does not regularize the linear terms in the ReLU network, which is not standard practice. I think that the results will be much stronger if the authors either:
    + Supply some optimization result that motivates studying this setting.
    + Extend the results to the multivariate setting using the results from Ongie et al (2019).


**Summary Of The Paper:**

The paper studies the implicit bias of univariate one-hidden layer ReLU networks with infinitesimal weight decay. In this setting, the authors derive a geometric description for all the networks that interpolate some given training data and attain the minimal value of the $\ell_2$ norm of the neuron’s weights. Then, the authors use this result to obtain a generalization guarantee for learning $1d$ Lipschitz functions.

**Summary Of The Review:**

Overall, I think that the paper's contributions are not sufficient for acceptance, as discussed under “weaknesses”. Supplying some additional results, as suggested above, will significantly strengthen the paper.

---

> ### Author Response · Authors · 2022-11-07
> **Thank you + Reply regarding generality of results**
>
> #### **Summary**
>
> We thank the reviewer for their careful reading of our paper. The main issue raised by the reviewer is the generality of our results, which the reviewer deems as insufficient for two reasons:
>
> * the input dimension equals 1
> * there is no accompanying optimization guarantees
>
> We agree with the reviewer that it would have been nice to extend our results to these two settings (and we wrote this in the discussion of limitations and future work section). However, ***we respectfully ask the reviewer to reconsider whether the results, as they are, are sufficient for publication, in light of the two reasons below.***
>
> #### **Reason 1: Requiring Optimization and Generalization in One Paper is Too High of a Bar**
>
> Requiring results about even somewhat realistic neural networks to combine generalization and optimization guarantees is almost certainly too high a bar for publication. Indeed, this would exclude vast swathes of influential work on neural network theory, such as
>
> * Work on complexity measures such as VC dimension and Rademacher complexity, which give good test error guarantees only under the assumption that one can (approximately) minimize an ERM-type objective. These article rarely take up the question of how whether a given optimization algorithm can actually find an approximate loss minimizers with high probability. Our article is very much of this flavor and gives, to our knowledge, the first even mildly realistic generalization bounds for learning Lip 1 functions.
>
> * Literature on the NTK, which proves that, with high probability, gradient-based optimization is successful in sufficiently wide neural networks from a reasonable init with MSE loss. With a few rare exceptions, this style of analysis, with probably several hundreds of theoretical papers in NeurIPS, ICML, ICLR, etc, says nothing about generalization. This is because these works typically show that one can interpolate even a pure noise dataset. Indeed, one of the main motivations for our results is to understand how the structure in the training dataset corresponds to the simplicity of predictions, if optimization were successful.
>
>
> #### **Reason 2: Extensions to Higher Input Dimension are Technically and Conceptually Challenging**
>
> Minimal $\ell_2$-norm interpolation for a training dataset D with one dimensional inputs has several simplifying features, which are very much not present even in dimension 2:
>
> * When the input dimension equals $1$, it is simple to identify at least one element in RidgelessReLU(D), namely the connect-the-dots interpolant $f_D$. Thus, one can directly compute the value of the minimal $\ell_2$-norm needed to interpolate D. In input dimension $2$ and higher, it is not known how to construct even one example of a function in RidgelessReLU(D)!
>
> * In input dimension $1$, the effect of each neuron $x\mapsto \mathrm{ReLU}( a x + b )$ is almost local. More precisely, its effect on the jump in the gradient of the ReLU network is localized to just its breakpoint $x=-b/a$. This property that allows one to represent any continuous piecewise linear function from $\mathbb R$ to $\mathbb R$ as a one layer ReLU network. However, in input dimension $2$ and higher, it is decidedly false that every continuous piecewise linear function can be represented by a one layer ReLU network (even if the discontinuities for the gradient are along straight lines). This stems from the fact that a neuron $x\mapsto \mathrm{ReLU}( a x + b )$ turns on and off along an entire co-dimension $1$ hyperplane on inputs solving $ax+b=0$ and that the jump in the gradient is constant everywhere along this hyperplane.

---

### Official Review · Reviewer_izUL · 2022-10-30

**Confidence:** 4
**Correctness:** 3
**Technical Novelty And Significance:** 4
**Empirical Novelty And Significance:** Not applicable
**Recommendation:** 5

**Clarity, Quality, Novelty And Reproducibility:**

The main part of the paper is quite clear and well-motivated. But the proof of the main theorem 1.3 is somewhat hard to piece together. It seems scattered through the Appendix. There needs to be a better organization of this - maybe a summary of the proof in the main paper which walks the reader through the steps.

**Strength And Weaknesses:**

The framing of the question is quite interesting and its generalization implications are intriguing. The paper zones into a very special case and solves it in details. The approach is very appreciable.

The weakness of the paper is in its organization of the main proof.

**Summary Of The Paper:**

This paper asks a very interesting question about implicit bias of neural training - in particular about interpolating depth 2 nets. Instead of trying to ask which of these interpolating nets would some training algorithm find, this paper tries to come up with a geometrical description of a subset of these nets which have a certain minimum norm condition on its parameters - when the input and the output dimension of the net is 1.

Also the paper chooses to work with an unconventional class of nets which have an additional linear unit. But then one is left to wonder why the authors' didn't frame it as a result about R -> R depth-2 ResNets - which is what this looks like. There are a few  more ambiguities like this which I shall point out in the later parts of the review.

**Summary Of The Review:**

Its a very nice piece of work and I think it can lead to some very interesting further work. My main complaint is with the writing/organization of the main theorem's proof - and some claims which seem to have feet of clay - like the following line at the end of page 3,

"The elements of RidgelessReLU(D) can intuitively be thought of as all univariate one layer ReLU networks that minimize a weakly penalized loss...and the strength lambda of the weight decay penalty is infinitesimal."

Either this is a claim that the authors should be be able to put in the proof for - or they should formally state this as a conjecture.
Currently the wording is quite dodgy.

---

> ### Author Response · Authors · 2022-11-07
> **Thank you + Proposed Changes**
>
> ### **Summary**
>
> We thank the reviewer for their careful reading of the paper and overall positive assessment of the results. We also thank the reviewer for their actionable suggestions for improvement. We are currently working on a revision to address them as follows and **respectfully invite the reviewer to comment on whether they address the reviewer's primary concerns:**
>
> ### **Reviewer Comment 1: Heuristic Explanation of RidgelessReLU(D)**
>
> We understand why the reviewer is not happy with the precision in the sentence “The elements of RidgelessReLU(D) can intuitively be thought of as all univariate one layer ReLU networks that minimize a weakly penalized loss...and the strength lambda of the weight decay penalty is infinitesimal." Our use of the word ‘intuitive’ was meant to convey that this is a rough but not rigorous (at least not made rigorous in the current paper) way to think about RidgelessReLU(D). In the revised version of the paper, we will clarify this point, explicitly saying that this is a heuristic only meant to aid a reader in motivating the definition of RidgelessReLU(D) and that we do not try to prove it rigorously.
>
> ### **Review Comment 2: Outline of Proof**
>
> We agree with the reviewer that adding an outline of the main steps of the proof is a good idea. In the revision, we will add a new section doing this just before the ‘Discussion of Limitations and Future Work’. The key steps are the following:
>
> * “Local Straightening” given in Proposition A.1. This shows that any element f in RidgelessReLU(D) must be either convex or concave on any interval of the form $(x_i, x_{I+1})$ between two consecutive inputs in the training data.
> * “Linearity at Endpoints” given in Proposition A.3. This shows that any element f in RidgelessReLU(D) agrees with $f_D$ to the left of $x_2$ and to the right of $x_{m-1}$
> * “Left-right Compatibility” given in Propositions A.4, A.5, A.6. This gives constraints, by dividing into cases, on the monotonicity of “incoming slopes” $s_{in}(x_i)$ and “outgoing slopes” $s_{out}(x_i)$ of any f in RidgelessReLU(D).
>  * A corollary of the preceding results is that RidgelessReLU(D) is a subset of the set of functions satisfying the conclusions of Theorem 1.2.
>  * Finally, Proposition A.7 the set of functions satisfying the conclusions of Theorem 1.2  are a subset of RidgelessReLU(D)
>
> ### **Reviewer Comment 3: Overall Clarity**
>
> Beside the two points above, we would like to indicate our respectful disagreement with the reviewer’s statement “some claims … seem to have feet of clay,” which to us seems to imply that there are serious problems with the proofs themselves. We honestly attempted to write the proofs to be at least locally transparent, with all details carefully laid out and 7 figures in the appendix to assist the reader in understanding the arguments and the point of the main statements.

---

> > ### Comment · Reviewer_izUL · 2022-12-10
> > **The set RidgelessReLU(D) needs more explanations.**
> >
> > Thanks for the comments.
> > I am still barely convinced about the setup here.
> >
> > Let me ask a few more clarifying questions,
> >
> > 1.
> > Since the space of all parameters is being kept unbounded, why do we know that any $\theta$ at all exists where the condition $C(\theta)=C_*$ can be satisfied?
> >
> > 2.
> > Even if we assume that the above exists, why is the set RidgelessReLU(D) a non-empty set?
> > Why is there at all an interpolant at which the equation $C(\theta)=C_*$ is being satisfied?
> >
> > 3.
> > Isn't it true that the main result in this paper is a structural result about this set? So there is no relationship that has been proven between this set and the output of any algorithm, right? Then why is this any establishing of ``implicit basis" if no conclusion is being proven about the output of any optimization algorithm in the presence of multiple solutions?

---

> > > ### Author Response · Authors · 2022-12-12
> > > **response to questions**
> > >
> > > We thank reviewer izUL for their questions and respond as follows:
> > >
> > > **$C(\theta)= C_*$ can indeed be achieved at finite $\theta$:**
> > >
> > > Theorem 1.3 (already proved in prior work of Saverese et. al.) shows that the ``connect-the-dots'' interpolant $f_{\mathcal D}$ belongs to $\mathrm{RidgelssReLU}(\mathcal D)$. Thus,
> > >
> > > $$
> > > C_* = ||D_{TV}f_{\mathcal D}|| = \text{ sum of absolute value of jumps of slopes in }f_{\mathcal D}
> > > $$
> > >
> > > In particular, $C_*$ is finite, so only $\theta$ of bounded norm can possibly give rise to functions $z(x;\theta)$ in $\mathrm{RidgelssReLU}(\mathcal D)$. Moreover, $f_{\mathcal D}$ - like any other continuous and piecewise linear function from $\mathbb R$ to $\mathbb R$ with $m$ breakpoints - can be written exactly as a width $m$ ReLU network with a single linear unit.
> > >
> > > **The space $\mathrm{RidgelssReLU}(\mathcal D)$ is non-empty:**
> > >
> > > Theorem 1.3 (already proved in prior work of Saverese et. al.) shows that the ``connect-the-dots'' interpolant $f_{\mathcal D}$ belongs to $\mathrm{RidgelssReLU}(\mathcal D)$. Thus, $\mathrm{RidgelssReLU}(\mathcal D)$ is non-empty. A priori this is not obvious.
> > >
> > >
> > > **In What Sense Do We Address Implicit Bias?**
> > >
> > > The reviewer is right that there is no analysis of optimization in this article. However, many papers on implicit bias do not address optimization directly. Instead, and this our point of view, they seek to describe, among all functions that are computed by a given model and interpolate the training data, which ones can actually be obtained by a particular optimization scheme. From this point of view, our results show that if _any_ optimization algorithm returns, among all one layer ReLU networks that interpolate a given training dataset, one with minimum $\ell_2$-norm of the training data, then although there are often infinitely many such interpolants, they are automatically locally convex/concave as described by the main results.
> > >
> > > Thus, the sense in which we address implicit bias is that we identify how the structure in the training data interacts with the optimization choice of adding an infinitesimal $\ell_2$ penalty on weights to ensure that the resulting class of interpolants changes sign for its second derivative no more than required by the data. This strong constraint on the possible learned interpolants allows us to conclude that, uniformly over any such interpolants, ReLU networks can generalize on data generated by Lipschitz functions.

---

### Author Response · Authors · 2022-11-12
**Revision Uploaded**

We thank the reviewers again for their comments and suggestions. We have uploaded a revision addressing, as described in our response to each of the reviewers, their comments.

---

### Decision · Program_Chairs · 2023-01-20

**Decision:**

Reject

**Justification For Why Not Higher Score:**

See MetaReview.

**Justification For Why Not Lower Score:**

N/A

**Metareview: Summary, Strengths And Weaknesses:**

The paper characterizes solution concepts of a certain type of univariate one layer ReLU networks which interpolate. This paper has had quite a bit of discussion, and the reviewers and AC all feel that while achieving results in this area is difficult -- and they take the point that there are not many papers addressing both generalization and optimization -- the present paper still falls short of being a substantial contribution. There was also a discussion about whether the words "Implicit Bias" on the title are misleading. The AC is sympathetic to the extent that these words have been used in varied contexts, but they should probably be removed from this paper as it is talking about solution concepts rather than the behaviour of algorithms. Overall it was not felt that the contributions were of sufficient technical depth or of sufficient interest for a venue like ICLR after a substantial discussion amongst all of the reviewers.

**Summary Of Ac-Reviewer Meeting:**

See the summary in the meta-review for the summary. The AC went to the extent of having two different meetings due to timezone issues.